# Understanding the Behaviour of Human Cell Types under Simulated Microgravity Conditions: The Case of Erythrocytes

**DOI:** 10.3390/ijms23126876

**Published:** 2022-06-20

**Authors:** Cristina Manis, Alessia Manca, Antonio Murgia, Giuseppe Uras, Pierluigi Caboni, Terenzio Congiu, Gavino Faa, Antonella Pantaleo, Giacomo Cao

**Affiliations:** 1Department of Life and Environmental Sciences, Cittadella Universitaria di Monserrato, 09042 Monserrato, Italy; cristina.manis@outlook.it (C.M.); a-murgia@hotmail.it (A.M.); caboni@unica.it (P.C.); 2Department of Mechanical, Chemical and Materials Engineering, University of Cagliari, Piazza d’Armi, 09123 Cagliari, Italy; 3Department of Biomedical Science, University of Sassari, Viale San Pietro, 07100 Sassari, Italy; alessia_manca@hotmail.it; 4Department of Clinical and Movement Neurosciences, Institute of Neurology, University of College London, London NW3 2PF, UK; g.uras@ucl.ac.uk; 5Department of Medical Sciences and Public Health, University of Cagliari, Monserrato’s Campus, 09042 Monserrato, Italy; terenzio.congiu@unica.it (T.C.); gavinofaa@gmail.com (G.F.); 6Center of Advanced Studies, Research and Development in Sardinia (CRS4), Loc. Piscina Manna, Building 1, 09050 Pula, Italy; 7Sardinia AeroSpace District (DASS), at Sardegna Ricerche, Via G. Carbonazzi 14, 09123 Cagliari, Italy

**Keywords:** erythrocytes, microgravity, confocal and scanning electron microscopy, oxidative stress, mass spectrometry, lipidomics

## Abstract

Erythrocytes are highly specialized cells in human body, and their main function is to ensure the gas exchanges, O_2_ and CO_2_, within the body. The exposure to microgravity environment leads to several health risks such as those affecting red blood cells. In this work, we investigated the changes that occur in the structure and function of red blood cells under simulated microgravity, compared to terrestrial conditions, at different time points using biochemical and biophysical techniques. Erythrocytes exposed to simulated microgravity showed morphological changes, a constant increase in reactive oxygen species (ROS), a significant reduction in total antioxidant capacity (TAC), a remarkable and constant decrease in total glutathione (GSH) concentration, and an augmentation in malondialdehyde (MDA) at increasing times. Moreover, experiments were performed to evaluate the lipid profile of erythrocyte membranes which showed an upregulation in the following membrane phosphocholines (PC): PC16:0_16:0, PC 33:5, PC18:2_18:2, PC 15:1_20:4 and SM d42:1. Thus, remarkable changes in erythrocyte cytoskeletal architecture and membrane stiffness due to oxidative damage have been found under microgravity conditions, in addition to factors that contribute to the plasticity of the red blood cells (RBCs) including shape, size, cell viscosity and membrane rigidity. This study represents our first investigation into the effects of microgravity on erythrocytes and will be followed by other experiments towards understanding the behaviour of different human cell types in microgravity.

## 1. Introduction

Red blood cells (RBCs) are peculiar within the human body, given their unique physiological role as transporters of O_2_ and CO_2,_ with a specialized molecular apparatus for gas exchange [1]. In addition to this, RBCs present a differentiated membrane which allows them to be physically dynamic in terms of shape and structure in order to sustain the stresses within the blood flow [2].

RBCs have peculiar characteristics, such as absence of DNA, simplified structure and a metabolism that suggests that these cells have a special relationship with the external environment. They are very sensitive to their surroundings and also well equipped with antioxidant systems which essentially contribute to their function and integrity [1].

It is well known that the presence of gravity on Earth deeply influenced the evolution of living organisms [3]. During the past century, human space activities have opened new scientific challenges related to understand the effects caused by the low gravity environment (microgravity) on organisms at the cellular as well as systemic level [4]. Microgravity (µg) can regulate the behaviour and structural properties of cells and regarding RBCs effectively acts as modulator of their shape and function through various mechanisms [5]. Microgravity environment has resulted in modified transcriptome and cytoskeletal changes, resulting in both short and long-term morphological changes in the cells [6]. Propelled by the constant increase in the human presence within space environment such as the International Space Station (ISS), there has been increased attention on the effects of modelled microgravity on cells physiological functions.

The exposure to microgravity conditions during space flight is associated with various health risks, which potentially affect the body, making it vulnerable to secondary conditions such as the effects of radiation, physiological changes lysis, neocytolysis, muscle atrophy, hydrostatic pressure changes, and metabolic changes [7,8,9]. During spaceflight, astronauts are also subjected to a variety of stress factors, such as microgravity conditions and radiation, which lead to negative health consequences [10,11]. The main cause of cellular damage to astronauts is oxidative stress [12], which is caused by an imbalance between oxidant and antioxidant species [13,14]. Previous studies [15,16,17] have been conceived to report the most recent knowledge about band 3 protein, with specific regard to its functions in oxidative stress conditions, including oxidative stress-related diseases. The produced reactive oxygen species (ROS) are mainly superoxide anion (O^2−^), hydrogen peroxide (H_2_O_2_) and free radicals. An increase in oxidants causes an imbalance in favour of oxidizing species, which are the cause of several pathologies in humans, as well as being one of the causes of aging. In more detail, the astronauts are affected, on their return to Earth, by several health problems such as reduction in bone density up to 20%, muscle atrophy, endocrine, immune system and cognitive disorders in addition to cardiovascular dysfunctions [18]. In particular, blood samples from Russian cosmonauts have been investigated observing significant changes in phospholipids [19]. An increase in the percentage of phosphatidylcholine may be clearly associated with the increase in membrane rigidity. On the other hand, changes in physicochemical properties of the plasma membrane of erythrocytes (microviscosity and permeability) can influence the efficiency of oxygen transfer, the state of the haemoglobin and changes in the conformation of haematoporphyrin. Moreover, lipid and phospholipid compositions of erythrocyte membranes were assayed by thin layer chromatography followed by densitometric measurement of stained dots. This technique provides information on the entire lipid class, but hardly allows the recognition of the specific molecules that belong to that class, making it difficult to understand the biochemical mechanisms involved in achieving the final result [19]. However, most previous studies have investigated data derived after the end of a space flight, while only a few data related to changes occurring during a space flight are described in the literature. Concerning RBCs, it is known that spaceflight induces oxidative stress [20], although the direct role of microgravity is still unclear, while there are several pieces of evidence of blood homeostasis alterations such as the so-called space anaemia or Pseudopolycythemia [21]. During the early days of low gravity conditions, astronauts show a reduction in plasma volume and therefore an increase in erythrocytes relative volume [22,23]. The body’s adaptive response to this new condition consists of an increase in erythrocytes clearance and a decrease in erythropoietin synthesis [24]. Therefore, haemoconcentration occurs with an increase in the haematocrit that is favoured by the rapid decrease in erythrocytes selective counting and haemolysis [25].

On the basis of the considerations above, due to the high cost of space experiments, there is an increasing demand of the development of technologies to guarantee nearly zero-gravity environmental conditions on Earth [26].

To model microgravity conditions observed on the ISS, a g force between 10^−3^ and 10^−6^ has to be achieved, resulting in a weightless environment [1]. In order to do to so, international space agencies validated machines, namely random positioning machines (RPM) and 3D-clinostats, have been employed [1,27] (Figure 1). Such instruments simply produce a fast rotation within three axes, resulting in a null gravity [28,29].

Despite the great relevance of the topic, currently few studies have been carried out to investigate the behaviour of RBCs under simulated gravity conditions which demonstrated that these cells undergo a rapid sensing of the environment initially translated into a metabolic adaptive response and subsequently fixed in structural and morphological alterations, thus in modulations or alterations of functionality [1,30].

Since no other data on this subject are available in the literature, here we present a novel approach to understand the changes to RBCs under microgravity conditions, specifically related to their membrane components, using different biochemical and biophysical techniques, such as scanning electron microscopy (SEM) and confocal microscopy together with the exploitation of the great potential offered by chromatographic and mass spectrometry to comprehensively evaluate the structure and function of RBCs in normal and in 3D-clinostat-delivered microgravity conditions. These analyses collectively provide additionally insights into red cell morphology and membrane property changes in direct relation to biological mechanisms.

## 2. Results and Discussion

To understand how RBCs sense microgravity and transduce this stimulus into morphological modification, confocal and scanning electron microscopy were used. RBCs were clinorotaded at given time-points. Specifically, 0, 6, and 9 h were set up, while control RBCs were placed in the static bar at 1 g to undergo the same vibration of the samples under microgravity conditions. Figure 2 and Figure 3 allow us to visualize, by confocal and scanning electron microscopy respectively, the progressive changes in RBCs morphology related to microgravity condition. Clinorotated RBCs already at 6 h differ significantly in shape and size when compared to RBCs at 1 g. No significant alterations were in fact present at 1 g (Panels A, B, C of Figure 2 and Figure 3), since RBCs showed a biconcave shape, at 6 and 9 h being their conditions identical at T_0_. On the contrary, the microgravity-exposed RBCs showed, at increasing times (Panels E, F of Figure 2 and Figure 3) more severe morphological defects, probably due to a faster occurrence of structural weakening and alterations.

In fact, cells were in the form of echinocytes characterized by convex rounded protrusions or spicules evenly spaced around the cell circumference, with a sharp increase in the spiculed morphology at 9 h of clinorotation, usually considered as a more severe marker of cell senescence and corresponding oxidative stress.

These observations can be certainly correlated with the redox power (e.g., the GSH/GSSH ratio) and the intracellular and extracellular ATP which determine cell’s resources and the corresponding metabolic status [1]. Concerning RBCs, we are aware of space-flight-induced oxidative stress [20] in which the direct role of microgravity is still controversial. The relationship between metabolic adaptations induced by microgravity and their structural or morphological characteristics is still unclear.

It is worth mentioning that the observed landscape obtained by microscopy analysis was indicative of a progressive loss of the cellular function and defence, coupled with a degradation of RBCs’ structures. On the basis of these premises, we quantified oxidant and antioxidant species to better understand some of the RBCs’ metabolic strategies exerted under stress, and compared their time-dependent evolution under static or microgravity conditions. To analyse the oxidative stress experienced and the imbalances between antioxidant species by red blood cells during the performed experiments, we decided to evaluate ROS and malondialdehyde (MDA-TBARS) levels as markers of oxidative stress. In addition, total antioxidant capacity (TAC) and glutathione (GSH) were measured, as antioxidant defence species, which can prevent cellular damage caused by oxidant species such as ROS and MDA. The measurement of TAC allows one to estimate the ability of cells to counteract oxidative stress-induced damage in cells.

The evaluation of antioxidant and ROS species was carried out on plasma samples exposed to simulated microgravity (µg) and Earth gravity condition (1 g). Figure 4 Panel A shows the extracellular ROS level at different time point (T_0_, 30–60–90 min and 2–3–6–9 h). Samples exposed to simulated microgravity showed a constantly increase in ROS levels compared to the control samples at 1 g. After 30 mins, level of ROS increased by 50% in microgravity samples and up to 100% at 9 h. In Figure 4, Panel B the TAC expressed in µmol/L is reported. It is seen that clinorotated samples displayed a significative reduction, approximately by 60% of TAC concentration at 2 h as compared to control samples which remains fairly constant. Figure 4, Panel C shows the total GSH concentration expressed in µmol/L. Samples exposed to clinorotation display a significant and constant decrease approximately by 85% in concentration up to 9 h while a non-significant decrease was observed under 1 g condition. Figure 4, Panel D shows the MDA concentration expressed in µmol/L. MDA concentration was higher in samples exposed to microgravity condition.

These results provide a confirmation of remarkable changes in red cell cytoskeletal architecture and membrane stiffness due to oxidative damage, validated in our study by the marked increase in ROS and by a significant decrease in TAC. RBC’s antioxidant defence system become no longer able to reduce the levels of reactive oxygen species (ROSs). Progressive incubation under microgravity condition is also thought to be a consequence of oxidative damage due to the increase in MDA concentration and the decrease in GSH. All these factors contribute to the deformability of the RBCs including shape, size, cell viscosity, and membrane rigidity.

To evaluate the overall impact of simulated microgravity on membrane modifications, we decided to investigate the lipid profile of erythrocytes under normal and microgravity conditions by LC-QTOF-MS. The unsupervised principal component analysis (PCA) is able to extract meaning information from data without training a model on labelled data. The PCA model for the data acquired in positive (PCA R^2^X = 0.67, Q^2^ = 0.54) and negative ionization mode (PCA R^2^X = 0.432 and Q^2^ = 0.195), respectively, did not indicate any clusters related to clinorotation. However, the arrangement of the samples in the multivariate space appeared to be influenced by the time factor. To further limit the time factor influence, for each time point, we performed a Partial Least Square-Discriminant Analysis (PLS-DA). The validation parameters of the positive ionization acquisition (PIA) model built for the samples collected at 6 h showed a good classifying power and a good predictive ability (R^2^X = 0.729; R^2^Y = 0.856; Q^2^ = 0.514), while the latter one decreased in PLS-DA of samples collected at 9 and 24 h (R^2^X = 0.699; R^2^Y = 0.432; Q^2^ = 0.165 and R^2^X = 0.602; R^2^Y = 0.678; Q^2^ = 0.107), respectively. Due to the greater predictive ability of the PLS-DA model relating to samples collected after 6 h of experiment, we focused on metabolites that discriminate the class of clinorotated erythrocytes from those ones at 1 g conditions. Finally, an OPLS-DA was performed to find the discriminant metabolites. For the positive and negative ionization mode the OPLS-DA score plot reported in Figure 5a,b, respectively, the following parameters were obtained: R^2^X = 0.755; R^2^Y = 0.998 and Q^2^ = 0.409 R^2^X = 0.432 and Q^2^ = 0.195, respectively. Among the most discriminant compounds, resulting by PIA OPLS-DA model, the membrane phosphocholines PC16:0_16:0 and PC 33:5 were found to be upregulated in erythrocytes clinorotated for six hours when compared with control samples. The significance of the discriminating character of these metabolites was assessed on the basis of the VIP value, which was 1.1089 and 1.8294, respectively. On the other hand, from the negative ionization acquisition (NIA) OPLS-DA model, the phosphocholine PC18:2_18:2, PC 15:1_20:4 and sphingomyelin SM d42:1 (VIP values: 1.1197, 1.4849, and 1.5191) were upregulated in RBCs clinorotated for six hours when compared with control samples.

Considering the results of the analysis and the general literature on the topic, it can be concluded that RBCs are very sensitive to their surroundings, changing shape and reacting to their environment. In an ideal situation, the RBC exists as a biconcave disc, of which the structure, shape, and modifications to physical stress are defined by the cytoskeletal architecture supporting the plasma membrane [31,32]. Modifications in the cytoskeletal proteins involved in structural function alter the RBCs spectrin network and membrane integrity when re-shaping in response to an induced stress [33,34]. Dysfunctional responses to physical change can result in a loss of function in RBCs, recapitulating a disease-like scenario [35].

Recently, several studies have shown an upregulation of phosphocholines class in different cell types such as epidermal stem cells [36] or gastric cancer cells subjected to simulated gravity conditions [37]. The same studies were performed on blood samples from Russian cosmonauts by Ivanova et al. [19]. In this case, the research team observed that during the voyage there was an increase in the percentage of phosphatidylcholine lipid class associated with the increase in membrane rigidity. In fact, an increase in phosphatidylcholine is associated with an increase in the rigidity of the cell membrane. The specific composition of the membrane provides the RBCs a certain fluidity, necessary to modulate cell functions, while loss of such fluidity affects normal functions and cellular signalling. From the analysis of our samples, it appears that the 42: 1 sphingomyelin was upregulated in erythrocytes cultured under simulated microgravity conditions. Sphingomyelins are often associated with inflammation processes. In this case, the latter ones could be triggered by the known increase in ROS. Furthermore, the upregulation of phosphocholines substantially affects the phosphocholines PC 18: 2_18: 2 and PC 15: 1_20: 4. Both of these phosphocholines have different unsaturation, considerably increasing the possible rigidity of the cell membrane, in particular this effect occurs within the first six hours of clinorotation.

## 3. Materials and Methods

### 3.1. Cell Culture

Freshly drawn blood (Rh+) from healthy adults of both sexes was used. Patients provided written, informed consent in the ASL. 1-Sassari (Azienda Sanitaria Locale.1-Sassari) centre before entering the study. This study was conducted in accordance with Good Clinical Practice guidelines and the Declaration of Helsinki. No ethical approval was requested as human blood samples were used only to perform in vitro experiments. Blood anti-coagulated with heparin was stored in citrate-phosphate-dextrose with adenine (CPDA-1) prior to use. RBCs were separated from plasma and leukocytes by washing three times with phosphate-buffered saline (127 mM NaCl, 2.7 mM KCl, 8.1 mM Na_2_HPO_4_, 1.5 mM KH_2_PO_4_, 20 mM HEPES, 1 mM MgCl_2_, and pH 7.4) supplemented with 5 mM glucose (PBS glucose) to obtain packed cells.

### 3.2. Microgravity Simulation

To verify whether RBCs could be affected by microgravity conditions, experiments were performed using a 3D random positioning machine (RPM, Fokker Space, Netherlands) at the laboratory of the Department of Biomedical Sciences, University of Sassari, Sardinia, Italy. The 3D RPM is a micro-weight (microgravity) simulator based on the principle of ‘gravity-vector-averaging’, built by Dutch Space. The 3D RPM is constituted by two perpendicular frames that rotate independently. The direction of the gravity vector is constantly changed so that the average of the gravity vector simulates a microgravity environment. The 3D RPM provides a simulated microgravity less than 10^−3^ g. The dimensions of the 3D RPM are of 1000 × 800 × 1000 mm (length × width × height). The 3D RPM is connected to a computer and through a specific software the mode and speed of rotation were selected. Random Walk mode with an 80 degree/sec (rpm) was chosen.

To simulate the effect of all the operating conditions, the following procedure was adopted.

Two millilitre tubes were carefully filled with packed RBCs and PBS-glucose (30% haematocrit) without air bubbles to avoid shearing of the fluid, in a dedicated room at 37 °C. Controls were placed in the static bar at 1 g to undergo the same vibration of the sample under µg conditions. Different time points were set (0–6–9–12–24–36–48 h). Subsequently, RBCs were centrifuged and resuspended in 1 mL of lysis buffer [5 mM Na_2_HPO_4_, 1 mM EDTA (pH 8.0)] and stored at −20 °C until use for further characterizations.

### 3.3. Confocal Microscopy Analysis

Once the experiments under micro and normal gravity conditions were performed at 0, 6, and 9 h, RBCs, after being fixed for 5 min in 0.5% acrolein in PBS, samples were rinsed three times, then permeabilized in PBS containing 0.1 M glycine (rinsing buffer) plus 0.1% Triton X-100 for 5 min and rinsed again 3× in rinsing buffer. To ensure complete neutralization of unreacted aldehydes, the cells were then incubated in rinsing buffer at room temperature for 30 min. After incubation, all nonspecific bindings were blocked by incubation for 60 min in blocking buffer (PBS containing 0.05 mM glycine, 0.2% fish skin gelatin and 0.05% sodium azide). Staining was performed using specific antibodies diluted in blocking buffer. After labelling, RBCs were allowed to attach to cover slips coated with poly-l-lysine, which were then mounted onto glass slides using Aqua-Mount (Lerner Laboratories, New Haven, Connecticut, USA). Samples were imaged with a Leica TCS SP5 confocal microscope equipped with a 60 × 1.4 numerical aperture oil immersion lens.

### 3.4. Scanning Electron Microscopy (SEM) Characterization

For three-dimensional SEM imaging and a fine resolution of erythrocytes structure, samples were fixed with 1% paraformaldehyde in 0.1 M Na-cacodylate buffer (pH 7.4) for 1 h at room temperature. After washing in Na-cacodylate buffer, samples have been decanted on microscope slides overnight. After numerous washes in PBS (Phosphate Buffered Saline), slides were dehydrated in ascending ethanol scales, dried at critical point in CO2 and then mounted on aluminium support (stub) with double-sided tape. Finally, samples were covered with a gold conductive film and observed at SEM (ZEISS SIGMA 300, Wetzla, Germany).

### 3.5. Oxidative Stress Analysis

Oxidative stress analyses were performed in lysate red blood cells and/or plasma according to the manufacturer’s instructions. Total antioxidant capacity (Cayman Antioxidant Assay kit 709001, Cayman Chemical, 1180 East Ellsworth Road Ann Arbor, Michigan 48108 USA) [38,39], reduced glutathione (GSH) (Cayman glutathione assay kit 703002, Cayman Chemical) [40,41], and thiobarbituric acid reactive substance (TBARS) (Cayman TBARS assay kit 10009055, Cayman Chemical, 1180 East Ellsworth Road Ann Arbor, Michigan 48108 USA) [38,39] were evaluated. Reactive oxygen species (ROS) level analysis was performed accordingly with method described by Tsamesidis et al. [14].

### 3.6. UHPLC-QTOF-MS Analysis

Prior to mass spectrometric analysis, 50 µL of human erythrocytes were extracted following the Folch procedure using 0.7 mL of a methanol and chloroform mixture (2/1, *v/v*). Samples were vortexed every 15 min up to 1 h, when 0.35 mL of chloroform and 0.15 mL of water were subsequently added. The solution was centrifuged at 12,000 rpm for 10 min, and 0.4 mL of the organic layer was transferred into a glass vial and dried under a nitrogen stream. The dried chloroform phase was reconstituted with 50 μL of a methanol and chloroform mixture (1/1, *v/v*) and 75 μL isopropanol:acetonitrile:water mixture (2:1:1 *v/v*). Quality control (QC) samples were prepared taking an aliquot of 10 μL of each sample. All samples thus prepared were injected in UHPLC-QTOF-MS/MS and acquired in negative ionization mode, while for positive ionization mode they were diluted in ratio 1:10. The chloroform phase was analysed with a LC-QTOF-MS coupled with an Agilent 1290 Infinity II LC system. An aliquot of 4.0 μL from each sample was injected in a Luna Omega C18, 1.6 μm, 100 mm × 2.1 mm chromatographic column (Phenomenex, Bologna, Italy). The column was maintained at 50 °C at a flow rate of 0.4 mL/min. The mobile phase for positive ionization mode consisted of (A) 10 mM ammonium formate solution in 60% of water and 40% of acetonitrile and (B) 10 mM ammonium formate solution containing 90% of isopropanol, 10% of acetonitrile. In positive ionization mode, the chromatographic separation was obtained with the following gradient: initially 80% of A, then a linear decrease from 80% to 50% of A in 2.1 min then at 30% in 10 min. Subsequently the mobile phase A was again decreased from 30% to 1% and was maintained at this percentage for 1.9 min and then brought back to the initial conditions in 1 min. The mobile phase for the chromatographic separation in the negative ionization mode differed only for the use of 10 mM ammonium acetate instead of ammonium formate. An Agilent jet stream source which was operated in both positive and negative ion modes with the following parameters: gas temperature, 200 °C; gas flow (nitrogen) 10 L/min; nebulizer gas (nitrogen), 50 psig; sheath gas temperature, 300 °C; sheath gas flow, 12 L/min; capillary voltage 3500 V for positive and 3000 V for negative; nozzle voltage 0 V; fragmentor 150 V; skimmer 65 V, octapole RF 7550 V; mass range, 50−1700 m/z; capillary voltage, 3.5 kV; collision energy 20 eV in positive and 25 eV in negative mode, mass precursor per cycle = 3. High purity nitrogen (99.999%) was used as a drift gas with a trap fill time and a trap release time of 2000 and 500 µs, respectively.

### 3.7. Multivariate Data Analysis

Mass spectrometric data acquired were pre-processed with the software MassHunter Workstation suite (Agilent Technologies, Santa Clara, California, USA). This software (Mass Profiler 10.0) allowed us to perform mass deconvolution, yielding a matrix containing all features present across all samples. This matrix was further processed with a pipeline based on the KNIME analytic platform [42], KniMet for the post-processing of metabolomics MS-based data [43]. Features were filtered based on their presence in QC samples (threshold = 40%) and the remaining features were collected in a data matrix subsequently processed using SIMCA software 14.0 (Umetrics, Umeå, Sweden). First, a Principal Component Analysis (PCA) was carried out. This unsupervised statistical analysis allowed us to observe samples and variables distribution in the multivariate space based on their similarity and dissimilarity. This was followed by partial least square-discriminant analysis (PLS-DA) with its orthogonal extension (OPLS-DA).

## 4. Conclusions

In this work we investigated, using different approaches, the behaviour of human RBCs under simulated microgravity conditions. No significant changes were present at 1 g in different time points, while the performed investigations using SEM and confocal microscopy, ROS, TAC, GSH, and MDA analysis, and lipid profile evaluation demonstrated that samples exposed to simulated microgravity conditions exhibited remarkable changes in red cell cytoskeletal architecture and membrane stiffness.

This work represents the first step of our research group towards the understanding the behaviour of human cells under microgravity conditions. Work is underway not only to analyse additional cell types with particular emphasis towards those ones that might be more important for future human missions but also to involve institutions and researchers at the national and international levels to strengthen the cooperation in this crucial field of enquiry.

It is apparent that the implications of this research work and the corresponding future developments are related to the basic investigation of the intrinsic mechanisms underlying the behaviour of several human cell types under microgravity conditions which will be able to provide a useful contribution towards both the understanding and the potential prevention of classical health risks associated to the astronauts involved in deep space exploration scenarios.

## Figures and Tables

**Figure 1 ijms-23-06876-f001:**
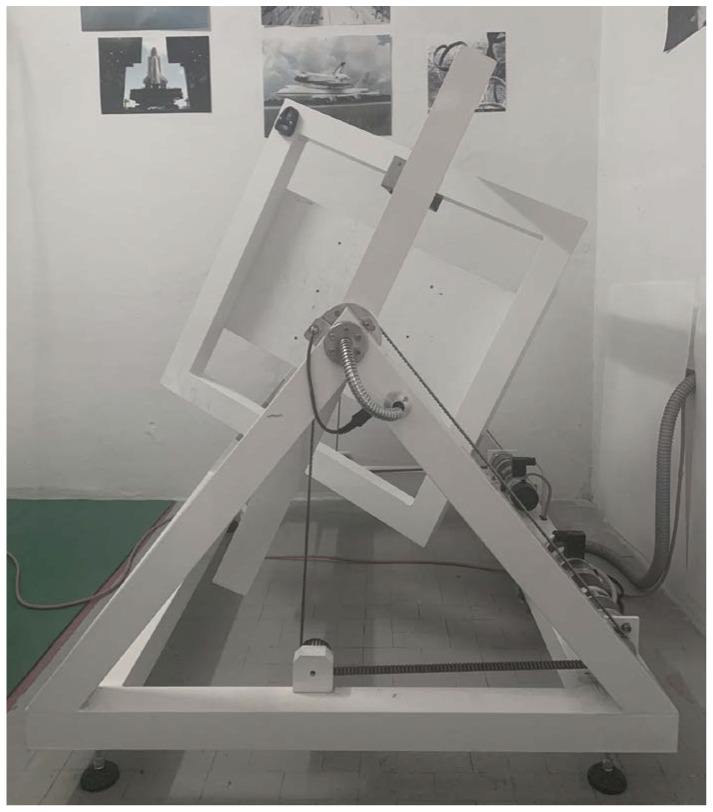
Random Positioning Machine (RPM, Fokker Space, The Netherlands).

**Figure 2 ijms-23-06876-f002:**
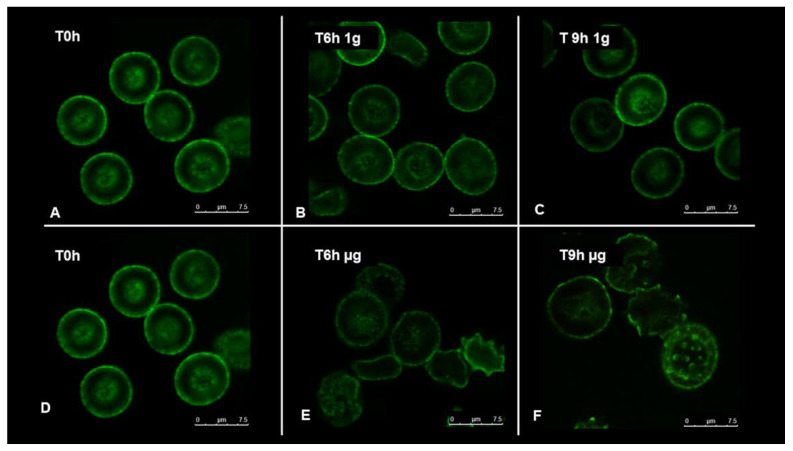
Confocal images of RBCs incubated for 6 and 9 h under terrestrial (1 g) (**A**–**C**) and microgravity conditions (µg) (**D**–**F**). Images were acquired using the same magnification with a Leica TCS SP5 X (Leica Microsystems, Wetzlar, Germany) confocal microscope equipped with a 60 × 1.4 numerical aperture oil immersion lens. The scale bar in the figure is 7.5 µm.

**Figure 3 ijms-23-06876-f003:**
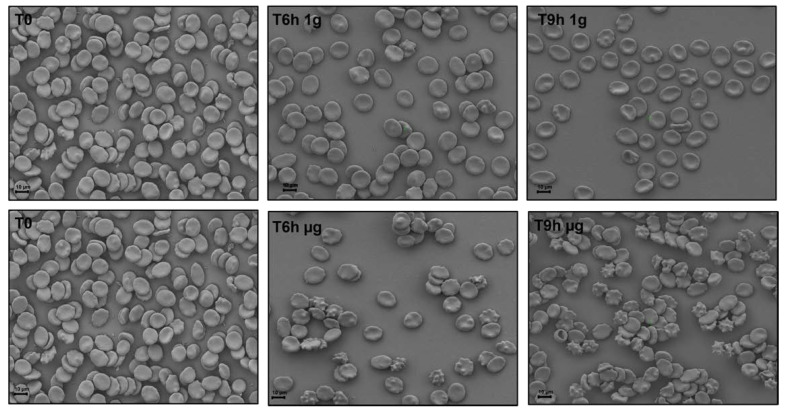
Scanning electron microscopy images of RBCs incubated for 6 and 9 h under terrestrial (1 g) and microgravity conditions (µg). Images were acquired using a SEM (ZEISS SIGMA 300, Wetzla, Germany). The scale bar in the figure is 10 µm.

**Figure 4 ijms-23-06876-f004:**
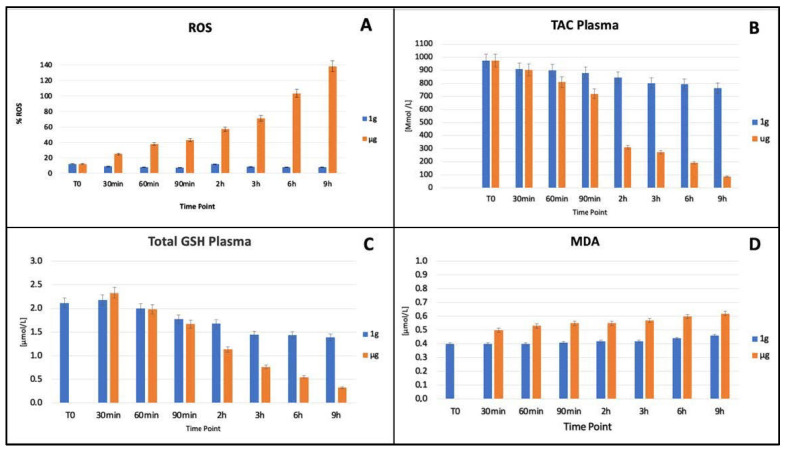
Evaluation of extracellular ROS level (**A**), TAC (**B**), GSH (**C**), and MDA (**D**) on plasma samples exposed to simulated microgravity (µg) and earth gravity conditions (1 g) at different time point (T0, 30–60–90 min and 2–3–6–9 h). Data are the average ± SD of three independent experiments.

**Figure 5 ijms-23-06876-f005:**
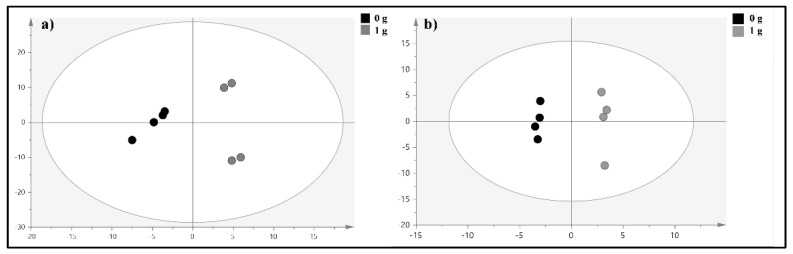
(**a**) OPLS-DA score plot for the PIA model and (**b**) OPLS-DA score plot for the NIA model. The grey circles represent the control samples, while black circles represent clinorotated erythrocytes samples.

## Data Availability

Written informed consent has been obtained from the patients to publish this paper.

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
