# Peer review of "Understanding the Behaviour of Human Cell Types under Simulated Microgravity Conditions: The Case of Erythrocytes"

_ijms, 2022, doi:10.3390/ijms23126876_

Round 1

Reviewer 1 Report

The study entitled “Understanding the behavior of human cell types under microgravity conditions: the case of erythrocytes“submitted to International Journal of Molecular Sciences by Manis C. and co-workers describes the changes that occur in the structure and function of red blood cells under simulated microgravity, compared to terrestrial conditions, at different time points using biochemical and biophysical techniques. The manuscript focuses an interesting topic that is worth to be published, but after minor revision.

Moderate editing of English language is required.

Could the authors provide more information in the abstract section the authors could better define the upregulation in PC16: 0_16: 0, PC 33: 5, PC18: 2_18: 2, PC 15: 1_20: 4 and SM d42: 1. Understanding is not immediate.

In the introduction it would be more appropriate to introduce the effect of oxidative stress on the red blood cell in general. For this purpose, you could consider the following manuscript:

https://doi.org/10.1002/jcp.30322;

Author Response

Manuscript ID ijms-1760990

Title: Understanding the behavior of human cell types under microgravity conditions: the case of erythrocytes

Answers to the reviewers

We would like to thank all the reviewers for the time spent on reviewing our manuscript. Appropriate changes were made and highlighted in the revised version of the manuscript according to the suggestions of reviewers and editor. Please find below a detailed point-by-point response to all comments (reviewers’ comments in black, our replies in red).

Reviewer #1

Moderate editing of English language is required.

The manuscript was edited for proper English language, grammar, punctuation, spelling, and overall style.

 Could the authors provide more information in the abstract section the authors could better define the upregulation in PC16: 0_16: 0, PC 33: 5, PC18: 2_18: 2, PC 15: 1_20: 4 and SM d42: 1. Understanding is not immediate.

We thank the reviewer for the suggestion. We added in the text a detailed chemical description of lipid membrane phosphocholines and sphingomyelin i.e. PC16: 0_16: 0, PC 33: 5, PC18: 2_18: 2, PC 15: 1_20: 4 and SM d42: 1. We changed the text accordingly at rows 31-32, 368, 372-373

 In the introduction it would be more appropriate to introduce the effect of oxidative stress on the red blood cell in general. For this purpose, you could consider the following manuscript: https://doi.org/10.1002/jcp.30322;

We thank the reviewer for highlighting this point. Thank you for the reference, which we have now cited in the revised manuscript.

We also have added in the text the general effect of oxidative stress (line 81) as follow:

Previous studies [15-17] has been conceived to report the most recent knowledge about band 3 protein, with specific regard to its functions in oxidative stress conditions, including oxidative stress-related diseases.

Reviewer 2 Report

1.       The meaning of the acronyms must be entered in the abstract. The abstract will be collected in the databases. Readers should be able to understand the abstract without having to read the article. In particular, the meaning of the following abbreviations should be entered. ROS, TAC, GSH; MDA, RBC

2.       As readers may not be familiar with machines that simulate microgravity, it would be interesting to include one or more pictures of the 3D RPM i device used in the manuscript.

3.       Please use the term microgravity consistently. Sometimes it appears as microgravity (μg) and sometimes as microgravity. It should be used in the same way throughout the manuscript.

4.       In line 241, they refer to some studies performed by Ivonova et al., but no bibliographic reference is provided. This should be corrected.

5.       The bibliography should be revised, correcting errors. As an example, Bibliographic reference number 3 is incomplete. The page number is missing. Adamopoulos, K.; Koutsouris, D.; Zaravinos, A.; Lambrou, G.I. Gravitational Influence on Human Living Systems and the Evolution of Species on Earth. Molecules 2021, 26, 2784

6.       In reference 6, the page number is also missing.

7.       The statistical analysis should be explained in more detail. In particular, it refers to machine learning when it talks about unsupervised principal component analysis. IT should be clarified because readers could be not familiar with the technique,

8.       It would be interesting to include a final paragraph commenting on the implications of this research and future developments in this line of research.

Author Response

Manuscript ID ijms-1760990

Title: Understanding the behavior of human cell types under microgravity conditions: the case of erythrocytes

Answers to the reviewers

We would like to thank all the reviewers for the time spent on reviewing our manuscript. Appropriate changes were made and highlighted in the revised version of the manuscript according to the suggestions of reviewers and editor. Please find below a detailed point-by-point response to all comments (reviewers’ comments in black, our replies in red).

Reviewer 2

  1. The meaning of the acronyms must be entered in the abstract. The abstract will be collected in the databases. Readers should be able to understand the abstract without having to read the article. In particular, the meaning of the following abbreviations should be entered. ROS, TAC, GSH; MDA, RBC

We thank the reviewer for the suggestion. The abstract has now been revised as suggested.

  1. As readers may not be familiar with machines that simulate microgravity, it would be interesting to include one or more pictures of the 3D RPM device used in the manuscript.

We thank the reviewer for highlighting this point. We have now added the picture of the 3D RPM device used for this study (Figure 1, pag 3)

  1. Please use the term microgravity consistently. Sometimes it appears as microgravity (μg) and sometimes as microgravity. It should be used in the same way throughout the manuscript

We thank for the suggestion. Manuscript has been revised using the term “microgravity” instead of μg

  1. In line 241, they refer to some studies performed by Ivonova et al., but no bibliographic reference is provided. This should be corrected.
  2. The bibliography should be revised, correcting errors. As an example, Bibliographic reference number 3 is incomplete. The page number is missing. Adamopoulos, K.; Koutsouris, D.; Zaravinos, A.; Lambrou, G.I. Gravitational Influence on Human Living Systems and the Evolution of Species on Earth. Molecules 2021, 26, 2784
  3. In reference 6, the page number is also missing.

We thank the reviewer for pointing out our mistakes. We have now added the reference 19, line 243, page 7; (no-tracked changes file)

We have also checked in detail that all references cited in text are also cited in the reference list, and vice versa. We have checked that the referencing follows the style currently used in IJMS

The statistical analysis should be explained in more detail. In particular, it refers to machine learning when it talks about unsupervised principal component analysis. IT should be clarified because readers could be not familiar with the technique.

 We thank the reviewer for the helpful suggestions. We clarify at rows 201 and 202 the meaning of unsupervised principal component analysis. We believe that in this mode the reader can understand the acronym. Now the sentence reports: “The unsupervised principal component analysis (PCA) is able to extract information from data without training a model on labeled data. The PCA model for the data acquired in positive (PCA R2X=0.67, Q2= 0.54) and negative ionization mode (PCA R2X=0.432 and Q2=0.195), respectively did not indicate any clusters related to clinorotation.” (lines 201-205; page 6) (no tracked changes file)

  1. It would be interesting to include a final paragraph commenting on the implications of this research and future developments in this line of research.

We thank the reviewer for this helpful suggestion.  We have now placed few sentences at the end of the conclusions: (Page 10, Lines 377-382) (No-tracked changes file). It is apparent that the implications of this research work and the corresponding future developments are related to the basic investigation of the intrinsic mechanisms underlying the behavior of several human cell types under microgravity conditions which will be able to provide a useful contribution towards both the understanding and the potential prevention of classical health risks associated to the astronauts involved in deep space exploration scenarios.
